# Hormonal Gut–Brain Signaling for the Treatment of Obesity

**DOI:** 10.3390/ijms24043384

**Published:** 2023-02-08

**Authors:** Eun Roh, Kyung Mook Choi

**Affiliations:** 1Division of Endocrinology and Metabolism, Department of Internal Medicine, Hallym University Sacred Heart Hospital, Hallym University College of Medicine, Anyang 14068, Republic of Korea; 2Division of Endocrinology and Metabolism, Department of Internal Medicine, Korea University College of Medicine, Seoul 02841, Republic of Korea

**Keywords:** gut hormones, hypothalamus, brainstem, energy metabolism, anti-obesity agents

## Abstract

The brain, particularly the hypothalamus and brainstem, monitors and integrates circulating metabolic signals, including gut hormones. Gut–brain communication is also mediated by the vagus nerve, which transmits various gut-derived signals. Recent advances in our understanding of molecular gut–brain communication promote the development of next-generation anti-obesity medications that can safely achieve substantial and lasting weight loss comparable to metabolic surgery. Herein, we comprehensively review the current knowledge about the central regulation of energy homeostasis, gut hormones involved in the regulation of food intake, and clinical data on how these hormones have been applied to the development of anti-obesity drugs. Insight into and understanding of the gut–brain axis may provide new therapeutic perspectives for the treatment of obesity and diabetes.

## 1. Introduction

The prevalence of obesity is increasing worldwide and has more than doubled in Korea over the past 20 years [1,2]. According to the results of the 2020 Obesity Fact Sheet published by the Korean Society for Obesity, the prevalence of stage 1 obesity (body mass index (BMI) 25–29.9 kg/m^2^) has increased by 1.12 times over the past 20 years, while that of stage 2 (BMI 30–34.9 kg/m^2^) and stage 3 obesity (BMI ≥ 35 kg/m^2^) has increased by 1.63 times and 2.79 times, respectively [3]. In particular, between 2007 and 2017, the prevalence of obesity in Korea significantly increased with a secular trend, especially in men [4]. Factors associated with significant weight gain over the past few decades include unhealthy dietary habits, including inadequate fruit and vegetable consumption, frequent consumption of energy-dense snack foods, meal skipping, and eating out, especially in younger adults [5]. Obesity is associated with increasing disease burden, including metabolic syndromes, type 2 diabetes, cardiovascular disease, nonalcoholic steatohepatitis, chronic kidney disease, site-specific cancers, musculoskeletal disorders, and premature death [6,7,8]. The cornerstone of body weight management is lifestyle-based therapy, which combines a tailored reduced-calorie diet, physical activity, and behavioral counseling [9]. Although most people with obesity attempt lifestyle interventions, only approximately 40% achieve clinically meaningful sustained weight loss (≥5% body weight) [10,11]. Pharmacological management of obesity as an adjunct to lifestyle intervention is recommended in people with a BMI ≥30 kg/m^2^ or 27–30 kg/m^2^ with at least one obesity-related complication, such as type 2 diabetes, hypertension, dyslipidemia, or sleep apnea [12]. Overall, most of the approved anti-obesity medications prior to semaglutide and tirzepatide have been shown to provide a modest placebo-subtracted weight reduction of 3–7% after 6–12 months of treatment, with the usage of some medications being quite limited due to their propensity for side effects [13]. Metabolic surgery is currently the most effective treatment for chronic weight management, with long-term benefits for survival, cardiovascular outcomes, and other complications [14,15,16]. It is recommended for patients with a BMI ≥35 kg/m^2^ or ≥30 kg/m^2^ and obesity-related comorbidities in Korea [17], but the scalability of surgical treatment is limited. Previous studies have suggested the implication of obesity with the reduced post-prandial response of glucagon-like peptide-1 (GLP-1), cholecystokinin (CCK), and peptide YY (PYY), as well as the unleashed post-prandial reaction of ghrelin [18]. Improvement of the post-prandial secretion of gut peptides may partially contribute to remarkable effects on weight loss and glucose control after metabolic surgery [19].

The role of gut hormones and gut–brain communication in the regulation of food intake has become a hot topic in recent years. The bidirectional interaction between the gut and central nervous system (CNS) plays a pivotal role in a wide range of physiological control mechanisms, including appetite, energy homeostasis, and glucose homeostasis [20,21,22]. The potential of these peripheral signals to provide novel targets for developing anti-obesity medications is particularly intriguing. The most recently approved GLP-1 receptor agonist, semaglutide 2.4 mg, has shown greater efficacy than the existing anti-obesity drugs and was recently approved for chronic weight management in adults with obesity [23]. Tirzepatide, a novel glucose-dependent insulinotropic polypeptide (GIP) and GLP-1 receptor (GLP-1R) agonist, resulted in greater weight reduction than selective GLP-1R agonists by affecting tissues not targeted by selective GLP-1R agonists, and by activating and integrating both GIP and GLP-1R signaling pathways in the brain [24]. Tirzepatide treatment in obese patients induced significant and sustained weight loss of up to approximately 20% [25]. The focus of this review is to provide an overview of the hormonal gut–brain signaling involved in the regulation of food intake.

## 2. Brain Regulation of Appetite

The hypothalamus and brainstem are critically involved in sensing metabolic signals that convey information about the body’s energy status (Figure 1). The integration of these signals triggers specific and coordinated physiological responses aimed at regulating energy homeostasis through the regulation of appetite and energy expenditure [26,27,28]. The hypothalamic arcuate nucleus (ARC) is located close to the neighboring median eminence, a circumventricular organ, which receives direct circulating hormonal and nutrient signals [29]. The following two subpopulations of ARC neurons express appetite-related neuropeptides: anorexigenic pro-opiomelanocortin (POMC) and orexigenic agouti-related protein (AgRP) neurons. Neuropeptide Y (NPY), another potent orexigenic peptide, is co-expressed with AgRP. Both POMC and AgRP/NPY neurons provide projections to melanocortin 4 receptor (MC4R)-expressing hypothalamic paraventricular nucleus (PVN) neurons. ARC POMC neurons decrease food intake and increase energy expenditure by releasing alpha-melanocyte-stimulating hormones (α-MSH), and subsequently activating MC4R signaling. The anorexic effect of POMC neurons is mediated in part by the action of serotonin through the 5HT-2C receptor on these neurons [30]. Conversely, ARC AgRP/NPY neurons inhibit melanocortin signaling by directly inhibiting POMC neurons and indirectly antagonizing α-MSH action on MC4R [31]. Loss-of-function mutations in MC4R in rodents increase food intake, reduce energy expenditure and lead to severe obesity [32]. MC4R mutations in humans are the most common cause of severe early-onset obesity (up to 6% of early-onset obesity cases before 10 years of age) [33], suggesting a crucial role of the central melanocortin system in the maintenance of energy homeostasis. NPY signals through Y receptors, a family of G-protein-coupled receptors. NPY-induced stimulation of food intake is mediated primarily by Y1 and synergistically by Y5 receptors [34,35]. NPY is required for rapid feeding responses, whereas AgRP through MC4R signaling is important for the delayed stimulation of feeding [36]. TPVN is an important integration site related to whole-body energy homeostasis, since it receives diverse afferent inputs from AgRP and POMC neurons in the ARC, and extra-hypothalamic regions such as the nucleus of the tractus solitarius (NTS) of the hindbrain [37]. Several neuronal subsets in the PVN synthesize and secrete neuropeptides that have a net catabolic action, including the thyrotropin-releasing hormone, corticotrophin-releasing hormone and oxytocin. The ventromedial hypothalamus (VMH) controls non-shivering thermogenesis in brown adipose tissue (BAT) through modulating sympathetic nervous system outflow [38].

The caudal brainstem is another key brain area involved in the regulation of food intake. Similar to the ARC, the NTS in the hindbrain is anatomically close to the area postrema (AP), another circumventricular organ. Thus, the NTS is ideally situated to receive and integrate humoral and neural signals. In addition, communication between the periphery and the brain is mediated through the vagus nerve afferent fibers, which project to the NTS. A vagotomy reduces meal size and meal duration, confirming that vagal afferents transmit meal-related signals to the brainstem [39]. Intimate communication occurs between the hypothalamus and brainstem, since the NTS receives extensive neuronal projections from the PVN and vice versa [40]. Similar to hypothalamic neurons, NTS neurons produce appetite-regulating GLP-1, NPY, and POMC. NTS POMC neurons show signal transducer and activator transcription 3 (STAT3) activation in response to peripheral leptin administration [41]. Thus, circulating hormones and nutrients can act on both the hypothalamus and brainstem to transmit metabolic signals to the brain [27].

Circulating hormones refer to peripheral signals that can directly affect ARC neurons’ activity by passing across the median eminence. Leptin and insulin are adiposity signals that circulate in proportion to the amount of stored fat and inform the brain about long-term energy storage. Leptin receptors are highly expressed in several brain regions, including the hypothalamus, and are activated by leptin. Mice that lack leptin and its functional receptor (LepRb), that is *ob/ob* and *db/db* mice, respectively, develop hyperphagia, obesity, and type 2 diabetes [42]. The primary site of leptin’s satiety effect is considered as the hypothalamic ARC [43]. Intracerebroventricular injection of leptin into the ARC decreased food intake and weight gain. In addition, leptin action in POMC neurons stimulates locomotion and prevents diabetes in morbidly obese, diabetic, and severely hypoactive leptin receptor-deficient *db/db* mice [44]. The Janus family of tyrosine kinase (JAK)-STAT signaling is the representative signaling pathway of leptin in the regulation of food intake and energy homeostasis. Leptin binds to its receptors and initiates signaling pathways through the activation of JAK and tyrosine phosphorylation of STAT [45,46]. Phosphorylated STAT3 promotes leptin-mediated transcriptional regulation of key appetite-regulating neuropeptides, including POMC, AgRP, and NPY. Leptin stimulates the transcriptional activity of POMC, whereas the dominant negative expression of STAT3 suppresses this activity, indicating that LepRb–STAT3 signaling mediates the effects of leptin on energy metabolism [47].

Apart from the homeostatic regulation of food intake, satiety is influenced by palatability. Hedonic eating refers to food consumption just for pleasure beyond hunger and negative consequences. In general, humans show a preference for palatable foods, which are typically rich in sugar and/or fat. Ghrelin and GLP-1 have opposite actions on eating behaviors; ghrelin reinforces food reward, whereas GLP-1 attenuates various palatable food-motivated efforts [48]. Neural circuits involved in hedonic eating include dopaminergic neuronal projections from the ventral tegmental area (VTA) of the midbrain to the nucleus accumbens (NAc) and neuronal networks that include VTA projections to the prefrontal cortex, amygdala, and hypothalamus. There is clear evidence that leptin also regulates hedonic appetites. Central leptin treatment in rodents suppresses the ability of sucrose and high-fat food to modulate a place preference [49,50]. Leptin receptors are present in the VTA, and direct injection of leptin into the VTA suppresses food intake by suppressing both the electrical activity of VTA dopamine neurons and dopamine release into the NAc [51].

## 3. Gut Hormones Regulating Food Intake

Gut hormones are key metabolic signals in gut–brain communication and act as short-term regulators of food intake [52,53]. GLP-1, GIP, CCK, PYY, and oxyntomodulin (OXM) are secreted in response to nutrients and induce satiety, whereas ghrelin is secreted in anticipation of nutrients and stimulates the appetite. Glucagon is secreted in response to nutrient-deprived conditions. Gut hormones play a fundamental role in the regulation of energy metabolism by acting on the vagus nerve or the brain. GLP-1 and GIP share common properties with incretins, but have different biological characteristics (Figure 2). The maturation of incretin biology has led to the development of anti-obesity drugs that potently activate GLP-1R and/or GIP receptors (GIPR), setting new and even higher standards for performance.

### 3.1. Enteroendocrine Cells

The gut is one of the largest hormone-producing organs in the human body. The enteoendocrine cells are located throughout the gastrointestinal tract and constitute only <1% of the intestinal epithelial cell population [54]. However, they have critical roles as an important component of the gut–brain axis [55]. There are at least 15 types of EEC that have been described and they produce more than 20 hormones that influence various processes, including insulin secretion, gut motility, and food intake. Previously characterized EEC families include I-cells that secrete CCK, K-cells that secrete GIP, and L-cells that secrete GLP-1 and PYY. Recent work has suggested that ECCs exhibit an overlap of hormonal expression, which reflects factors such as location along the gut and exposure to ingested nutrients [56].

### 3.2. GLP-1

GLP-1, a product of proglucagon cleavage, is mainly synthesized by intestinal L cells. Circulating GLP-1 levels increase following meal ingestion and decrease after fasting. GLP-1 exerts a strong incretin effect; that is, it stimulates glucose-dependent stimulation of insulin secretion via the GLP-1R expressed in pancreatic islets. GLP-1 also causes a reduction in plasma glucagon concentrations, delayed gastric emptying, appetite suppression via neuronal pathways, and decreased hepatic glucose production [57]. Moderately elevated GLP-1 concentrations have important effects on pancreatic β and α cells, while higher concentrations achievable with GLP-1 analogs slow down gastric emptying and reduce appetite and food intake [58,59]. GLP-1 inhibits food intake mainly via the activation of neural activity in ARC POMC neurons [60,61], as well as the activation of neurons in the NTS [62]. GLP-1R is expressed in key brain areas that control energy balance, including the hypothalamus and brainstem [63,64], and nuclei in the mesolimbic reward system, such as the VTA and NAc [65]. GLP-1R agonists suppress the desire for hedonic food intake by interacting with the mesolimbic system [66]. The peripheral administration of GLP-1R agonists causes it to reach the hindbrain via circulation or vagal afferents [62]. The anorectic effect of exogenous GLP-1 is attenuated in rodents or humans with vagotomy [67,68], suggesting that the vagal-brainstem-hypothalamic pathway plays a role in the effects of circulating GLP-1 on food intake. Chronic treatment with GLP-1R agonists significantly increased heart rate, but did not induce an alteration in the sympatho-vagal balance in patients with type 2 diabetes [69].

### 3.3. GIP

GIP is secreted from K cells of the upper small intestines and regulates blood glucose levels via its insulinotropic and glucagonotropic action on the pancreas [70]. GIP also plays an important role in lipid metabolism by promoting lipid storage by increasing adipose tissue blood flow and triglyceride uptake [71]. However, there is considerable uncertainty regarding how GIPR agonists or antagonists control energy metabolism. The genetic or pharmacological blockade of GIPR attenuates obesity development in rodents [72,73], whereas GIP overexpression reduces diet-induced obesity and improves glucose metabolism [74]. Recent studies have shown that the hypothalamic GIPR is a target for regulating energy balance [75,76]. GIPR is expressed in the hypothalamus, and activation of hypothalamic GIPR neurons reduces food intake [75]. Although central and peripheral administration of GIP stimulates neuronal activity in hypothalamic feeding centers, GIP fails to decrease food intake in mice with a CNS loss of GIPR [76]. In addition, the superior metabolic effects of GLP-1/GIP dual agonism compared with GLP-1 treatment alone are lost in mice with CNS GIPR deletions, indicating that the metabolic benefits of dual GLP-1/GIP agonists act partially via central GIPR signaling [76].

### 3.4. Glucagon

Glucagon levels fall after carbohydrate ingestion, and insufficient post-prandial glucagon suppression can contribute to the development of type 2 diabetes [77]. Glucagon decreases body weight through multiple biological actions, including the inhibition of food intake, stimulation of brown fat thermogenesis, and activation of lipolysis [78,79,80]. Glucagon’s anorectic action seems to be mediated via the liver–vagus–hypothalamus axis, since disconnecting the hepatic branch of the abdominal vagus blocks the ability of glucagon to inhibit food intake [81]. Long-term administration of glucagon in obese Zucker rats reduced body weight by up to 20% by increasing energy expenditure, without changing food intake [82]. The effects of glucagon on body weight and fat mass were partly via the hepatic expression of fibroblast growth factor 21 [83]. It seems obvious that glucagon affects body weight through a feeding-independent mechanism, most probably by stimulating energy expenditure and fat oxidation [84].

### 3.5. Ghrelin

Ghrelin, a stomach-derived peptide hormone, reaches the hypothalamus via the median eminence and acts as an orexigenic signal by activating NPY/AgRP neurons [85]. Circulating ghrelin levels, conventionally called the “hunger hormone”, increase in response to fasting and decrease after re-feeding. In humans and rodents with obesity, ghrelin levels are lower than lean controls [38]. Ghrelin exerts its biological effects on the energy balance through the growth hormone secretagogue receptor [86], which is expressed in AgRP neurons of the ARC [87]. The central and peripheral administration of ghrelin have been shown to increase food intake, weight gain, and adiposity in rodents [88,89]. There is evidence of an orexigenic effect of ghrelin in humans [90]. Ghrelin also stimulates hedonic eating by activating dopaminergic neurons in the VTA [91]. Vagal afferent neurons have ghrelin receptors and vagotomy abrogates the orexigenic effects of ghrelin administration [92]. Therefore, ghrelin seems to exert its effects via both endocrine mechanisms in the ARC and neuronal pathways to suppress vagal afferents [93]. Recently discovered liver-expressed antimicrobial peptide 2 (LEAP2), an endogenous antagonist of ghrelin, is suggested as a promising candidate for the treatment of obesity [94]. Further investigation is warranted, since ghrelin remains the only metabolic signal that potently activates the hypothalamic AgRP neurons that increase hunger.

### 3.6. Other Gut Hormones

Amylin is co-secreted with insulin from pancreatic β-cells and is highly homologous to calcitonin and calcitonin gene-related peptides (CGRP). It activates specific receptors, including the calcitonin G protein-coupled receptor. It reduces post-prandial glucagon secretion, slows gastric emptying, and reduces food intake [95]. The anorectic action of amylin is mediated by a direct effect on AP and subsequent activation of the central satiety pathway [96,97]. Amylin also affects the hedonic control of feeding by inhibiting feeding reward neurocircuits [98]. Combination therapy with long-acting analogs of amylin and the GLP-1 analog semaglutide exhibited potential additive effects on weight loss, suggesting their action on distinct neuronal subpopulations in the hindbrain [99].

PYY is co-secreted with GLP-1 from intestinal L-cells in response to nutrient ingestion [100]. Its major circulating form (PYY3–36) exerts a direct action on the hypothalamic ARC through Y2 receptor-mediated inhibition of NPY/AgRP neurons, hence the activation of POMC neurons [101,102]. Peripheral and intra-ARC administration of PYY3–36 reduces appetite and body weight by increasing neuronal activity in the ARC [102]. Consistent with the anorectic role of PYY, transgenic mice that lack or overexpress PYY exhibit opposite alterations in satiation and body-weight regulation [102,103]. Peripheral PYY3–36 also transmit satiety signals to the brainstem by increasing neuronal activity in the NTS via the vagal afferent pathway [104].

CCK is secreted from intestinal I-cells postprandially and binds to the CCK1 receptor (CCK1R) to decrease food intake by reducing meal size [105]. CCK1R is widely expressed in the vagal afferents, brainstem, and hypothalamus, and the satiety signals of CCK are transmitted to the NTS by vagal sensory neurons [106]. Fan et al. showed that CCK activates NTS POMC neurons and that brainstem MC4R signaling is required for CCK-induced appetite suppression [107]. In addition, hyperphagia in CCK1R null mice that ingest an HF diet is mediated by ghrelin [108], and the food-reducing effect of CCK is mediated by a functional synergistic interaction between leptin and CCK in the hypothalamic PVN [109]. Leptin-resistant diet-induced obesity (DIO) rats demonstrated decreased CCK sensitivity on vagal afferent nerves, which attenuates the effect of CCK on satiety [110]. In a randomized clinical trial using CCK-A receptor agonists, patients with overweight and obesity did not show a significant change in body weight or other cardiometabolic risk markers [111].

OXM is secreted from the intestinal L cells together with GLP-1 and PYY in the post-prandial state and exerts its anorectic action primarily through binding to GLP-1R and the glucagon receptor with lower affinity [112]. Since OXM has been proposed as a GLP-1R-biased agonist relative to GLP-1 [113], the GLP-1R-mediated effects of OXM differ from those of GLP-1. OXM decreases body weight by lowering food intake and increasing energy expenditure in both rodents and humans [114,115]. The administration of OXM increases neuronal activity in the hypothalamus, but not in the brainstem [116]. Manganese-enhanced magnetic resonance imaging studies have shown differential neural activation between OXM and GLP-1 in the hypothalamus [117], indicating that OXM may act via different hypothalamic pathways than those of GLP-1.

## 4. Clinical Applications of Gut Hormones for the Treatment of Obesity

Among Korean adults aged ≥30 years in 2018, the prevalence of diabetes was 13.8% and almost half of these adults (53.2%) reported obesity as a comorbidity [118]. Originally marketed for glycemic control in type 2 diabetes, GLP-1R agonists have been shown to be effective in reducing weight in individuals with and without type 2 diabetes (Table 1) [119]. In addition, combination treatment using both GLP-1R agonists and the sodium–glucose cotransporter 2 inhibitor (SGLT2i) is effective and tolerable in patients with type 2 diabetes [120]. Since SGLT2i has a different mechanism of action that inhibits glucose reabsorption in the renal proximal tubule and leads to glucosuria, combination therapy with GLP1RA and SGLT2i may produce additive metabolic and cardiovascular benefits [121]. Interestingly, GIP/GLP-1R co-agonists have been shown to be more effective at reducing body weight and blood glucose in obese mice than selective GLP-1R agonists [122]. Recent studies in rodents demonstrated that the appetite-modulating effects of brain GIPR signaling heralded the development of several different multi-receptor peptides that represent the latest advances in the treatment of diabetes and obesity [123].

### 4.1. Liraglutide

Liraglutide is a GLP-1R agonist marketed as Saxenda^®^ (1.8 mg daily subcutaneous injection) and Victoza^®^ (3 mg daily subcutaneous injection). Liraglutide has 97% amino acid sequence homology with native human GLP-1, differing primarily by substituting arginine for lysine at position 34 [124]. It is also created by attaching a C16 palmitoyl moiety with a γ-glutamic acid chemical spacer to the lysine residue at position 26 of the peptide precursor [124]. Liraglutide (Victoza) was approved by the Food and Drug Administration (FDA) in 2010 as an adjunct therapy to diet and exercise for the management of type 2 diabetes and at a higher dose (Saxenda) in 2020 for chronic weight management in patients with obesity or who are overweight with a BMI ≥ 27 kg/m^2^ and weight-related comorbidities. In the Satiety and Clinical Adiposity−Liraglutide Evidence (SCALE) Obesity and Prediabetes study, 3 mg of liraglutide daily, as an adjunct to diet and exercise, was associated with reduced body weight (−8.4 ± 7.3 kg in the liraglutide group compared with 2.8 ± 6.5 kg in the placebo group; a difference of −5.6 kg) and improved metabolic control for 56 weeks in non-diabetic obese patients [125]. Moreover, liraglutide treatment was associated with greater reductions in BMI, waist circumference, blood pressure, fasting lipid levels, and inflammatory markers than the placebo. Liraglutide, prescribed at 3.0 mg per day, also resulted in significantly greater weight loss over 56 weeks than the placebo among overweight and obese participants with type 2 diabetes [126] or obstructive sleep apnea [127]. In addition, the daily use of subcutaneous liraglutide (3.0 mg), compared with the placebo, maintained weight loss achieved by caloric restriction and induced further weight loss over 56 weeks [128]. The effects of liraglutide on cardiovascular outcomes were demonstrated in the Liraglutide Effect and Action in Diabetes: Evaluation of Cardiovascular Outcome Results (LEADER) trial that the treatment of liraglutide 1.8mg was associated with a 13% significant relative risk reduction in major adverse cardiovascular events in patients with type 2 diabetes and a high cardiovascular risk [129].

### 4.2. Semaglutide

Semaglutide is a GLP-1R agonist with a fatty diacid chain at lysine position 26, linked via a di-aminoethoxy γ-glutamic acid spacer, to improve albumin binding and protect against degradation by dipeptidyl peptidase-4 [130]. Semaglutide has 94% sequence homology with native GLP-1 and a longer half-life, allowing weekly administration [130]. Oral semaglutide has been developed as a permeation enhancer to increase solubility by increasing the pH of the local environment and using a passive transcellular route across cell membranes [131]. In addition, the oral delivery of GLP-1 through nanoparticles suggested better systemic and tissue bioavailability [132]. Semaglutide is marketed as Ozempic^®^ (subcutaneous injection, once-weekly dosing; available in 0.5; 1.0 mg dose) and Rybelsus^®^ (oral tablets, once-daily dosing; available at 3, 7; 14 mg doses). The FDA approved Ozempic in December 2017 and Rybelsus in September 2019 for use as an adjunct to diet and exercise to improve glycemic control in adults with type 2 diabetes. Semaglutide demonstrated superior reductions in hemoglobin A1c and superior weight loss (by 2.3–6.3 kg) compared with different comparators across the Semaglutide Unabated Sustainability in Treatment of Type 2 Diabetes (SUSTAIN) 1 to 5 trials [133]. In the SUSTAIN 6 trial, there were fewer first major adverse cardiovascular events with semaglutide with a 16% relative risk reduction [134].

In June 2021, the FDA approved the 2.4 mg weekly dose of semaglutide (Wegovy^®^), a once-weekly injection, for chronic weight management in adults with obesity or overweight with at least one weight-related condition, when used in combination with a healthy diet and exercise. The efficacy and safety of semaglutide as the sole weight loss medication were investigated in the Semaglutide Treatment Effect in People with Obesity (STEP) trials. Semaglutide 2.4 mg once weekly was associated with a significant weight reduction (−14.9% in the semaglutide group compared with −2.4% in the placebo group) during 68 weeks in overweight and obese participants without diabetes [135]. Weight loss with semaglutide was accompanied by greater reductions than the placebo in waist circumference, BMI, blood pressure, lipid levels, and inflammatory markers. STEP 4 showed that those who discontinued semaglutide treatment and switched to the placebo at week 20 regained weight of approximately 6 kg gradually [136]. In the STEP 8 trial, semaglutide had greater effects on body weight reduction than liraglutide among overweight and obese adults (−15.8% in the semaglutide group compared with −6.4% in the liraglutide group and −1.9% in the placebo group) [23]. In addition, concomitant treatment with semaglutide and a long-acting amylin analog, cagrilintide, induced more weight loss with acceptable safety, compared with semaglutide alone in a phase Ib trial [137]. Subsequent studies are needed to fully assess the efficacy and safety of a once-a-week fixed dose combination of CagriSema, comprising 2.4 mg semaglutide and 2.4 mg cagrilintide. As a class, GLP-1R agonists generally have a more favorable safety profile than several other anti-obesity agents. The most frequently reported adverse effects were mild-to-moderate gastrointestinal symptoms, such as nausea, vomiting, and diarrhea. Nausea is dose-dependent and limits the use of higher doses to induce greater weight loss [53,138]. As mentioned above, the underlying mechanisms are linked to gut–brain communication through the direct activation of the hypothalamus and hindbrain or indirect activation via the vagus nerve. Therefore, semaglutide reduced appetite and food intake and lowered the preference for fatty, energy-dense foods in obese patients [139].

### 4.3. Tirzepatide

Tirzepatide, a dual GIP and GLP-1R agonist, was developed by engineering GLP-1 activity into its GIP sequence [24]. Tirzepatide has equal affinity for GIPR compared with native GIP, but it binds to GLP-1R with approximately five-fold weaker affinity than native GLP-1 [24]. Tirzepatide was associated with significantly greater efficacy with regard to glucose control and weight loss compared with selective GLP-1R agonists, such as dulaglutide and semaglutide, and the beneficial effects were dose-dependent [140,141]. Mounjaro^®^ injection was recently approved by the FDA in May 2022 as an adjunct to diet and exercise to improve glycemic control in adults with type 2 diabetes. The SURMOUNT-1 trial is the first clinical trial to test the ability of tirzepatide to induce weight loss in overweight or obese patients [25]. During 72 weeks of treatment, participants taking tirzepatide at weekly doses of 5, 10, or 15 mg showed substantial weight loss with a mean weight change of 15.0%, 19.5%, and 20.9%, respectively, compared with only 3.1% in the placebo group [25]. Tirzepatide treatment also resulted in benefits with respect to changes in waist circumference, blood pressure, and lipid levels. The most common adverse events with tirzepatide were mild-to-moderate gastrointestinal events that occurred primarily during dose escalation and were similar to those reported in previous studies of incretin-based therapies. Treatment discontinuation was observed in 4.3%, 7.1%, 6.2%, and 2.6% of participants receiving 5, 10, and 15 mg tirzepatide doses and the placebo, respectively. Long-term outcome data are needed.

## 5. Conclusions

Obesity is one of the greatest health challenges worldwide and a global epidemic that causes many complications and chronic diseases. The significant morbidity and mortality rates associated with obesity have prompted a large amount of research to develop safe and effective anti-obesity medications. The bi-directional gut–brain axis, which controls hunger and satiety, is mediated by rapid-acting neuronal pathways and slow-acting hormonal mechanisms [142]. Energy homeostasis is regulated by complex and coordinated processes that involve multiple neural circuits. The hypothalamus and brainstem are the major brain regions responsible for the homeostatic regulation of energy homeostasis, receiving peripheral nerve and hormonal signals that convey information regarding energy availability. Gut hormones released in response to meals act as short-term regulators of food intake. These hormones, except for ghrelin, decrease appetite and food intake. Hormonal gut–brain signaling is mediated through afferent fibers of the vagus nerve that project to the NTS in the hindbrain or through circulation, reaching the brain through the median eminence of the hypothalamus or the AP of the brainstem.

As outlined in this review, recent clinical trials of advanced therapeutic candidates, including GLP-1R agonists, have provided hope that breakthrough pharmacological obesity treatments may be possible. Semaglutide 2.4 mg, a recently approved GLP-1R agonist, reduced body weight, with an average weight loss exceeding 10%. In addition, clinical trials of tirzepatide, a dual GIP/GLP-1R agonist, have reported a weight loss of up to 20%, providing new hope that obesity and its related co-morbidities can be managed with medication rather than surgical interventions. Cagrisema, a fixed combination of semaglutide and cagrilintide, is expected to provide encouraging clinical data. Unsurprisingly, the clinical results of tirzepatide treatment have spurred tremendous interest in GIP-based dual agonists and other combinatorial approaches (Table 2). Further clinical studies will determine whether medical treatments that can achieve comparable efficacy to metabolic surgery are available.

## Figures and Tables

**Figure 1 ijms-24-03384-f001:**
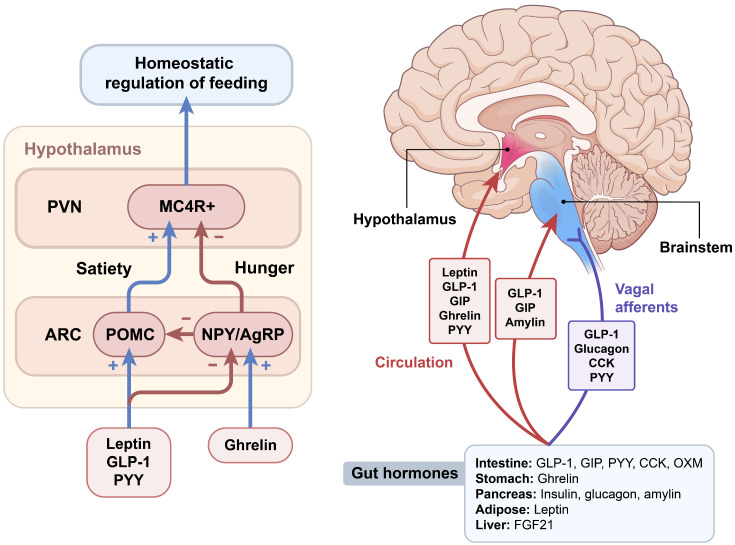
Gut–brain regulation of food intake. Various peripheral hormones, including gut hormones, regulate food intake by acting on integrated neural circuits in the hypothalamus and brainstem. The hypothalamic melanocortin system is a central hub for the regulation of homeostatic food intake, including appetite-inducing neurons that co-express neuropeptide Y (NPY) and agouti-related peptides (AgRP), and anorexic neurons that express pro-opiomelanocortin (POMC). Gut–brain communication is mediated through vagus nerve afferents that project to the nucleus tractus solitarius in the hindbrain, or via the circulation reaching the median eminence of the hypothalamus and area postrema of the brainstem. ARC, arcuate nucleus; CCK, cholecystokinin; FGF21, fibroblast growth factor 21; GIP, glucose-dependent insulinotropic polypeptide; GLP-1, glucagon-like peptide-1; MC4R, melanocortin 4 receptor; OXM, oxyntomodulin; PVN, paraventricular nucleus; PYY, peptide tyrosine tyrosine.

**Figure 2 ijms-24-03384-f002:**
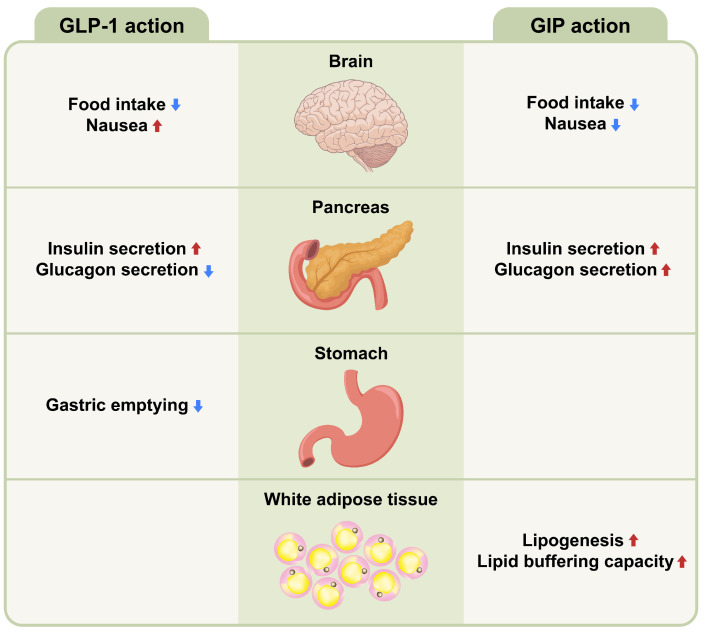
Metabolic action of GLP-1 and GIP on different tissues. Glucagon-like peptide-1 (GLP-1) and glucose-dependent insulinotropic polypeptide (GIP) share a function as incretins and have different pancreatic and extrapancreatic functions. GIP acts directly on the endocrine pancreas, brain and white adipose tissues, while GLP-1 acts directly on the endocrine pancreas, brain, and gastrointestinal tract. Up arrow (↑) symbol indicates increase, whereas down (↓) arrow symbol indicates decrease in the effect mentioned.

**Table 1 ijms-24-03384-t001:** Weight loss efficacy of liraglutide, semaglutide, and tirzepatie in overweight and obese patients without diabetes.

Trial Name	SCALE Obesity and Prediabetes	STEP 1	SURMOUNT-1
Drug tested	Liraglutide 3.0 mg SC daily	Semaglutide 2.4 mg SC weekly	Tirzepatide 5 mg, 10 mg, or 15 mg SC weekly
Study design	56-week, randomized, double-blind, placebo-controlled, 2:1 ratio	68-week, randomized, double-blind, placebo-controlled, 2:1 ratio	72-week, randomized, double-blind, placebo-controlled, 1:1:1:1 ratio
Study population	3731 overweight and obese patients without diabetes	1961 overweight and obese patients without diabetes	2539 overweight and obese patients without diabetes
Intervention	Liraglutide vs. placebo + lifestyle intervention	Semaglutide vs. placebo + lifestyle intervention	Tirzepatide vs. placebo + lifestyle intervention
Percent mean weight loss from baseline	Liraglutide −8.0% vs. placebo −2.6%, difference −5.4%	Semaglutide −14.9% vs. placebo −2.4%, difference −12.4%	Tirzepatide −15.0%, −19.5%, −20.9% with 5 mg, 10 mg, 15 mg vs. placebo −3.1%
Percentage of patients with ≥5% weight loss	Liraglutide 63.2% vs. placebo 27.1%	Semaglutide 86.4% vs. placebo 31.5%	Tirzepatide 85%, 89%, 91% with 5 mg, 10 mg, 15 mg vs. placebo 35%
Percentage of patients with >10% weight loss	Liraglutide 33.1% vs. placebo 10.6%	Semaglutide 69.1% vs. placebo 12.0%	Tirzepatide 69%, 78%, 84% with 5 mg, 10 mg, 15 mg vs. placebo 19%
Adverse events caused treatment discontinuation	Liraglutide 9.9% vs. placebo 3.8%	Semaglutide 7.0% vs. placebo 3.1%	Tirzepatide 4.3%, 7.1%, 6.2% vs. placebo 2.6%
Serious adverse effects	Liraglutide 6.2% vs. placebo 5.0%	Semaglutide 9.8% vs. placebo 6.4%	Tirzepatide 6.3%, 6.9%, 5.1% vs. placebo 6.8%

Abbreviations: SC, subcutaneous.

**Table 2 ijms-24-03384-t002:** A list of novel therapies for obesity and diabetes consisting of GLP-1-based multi-agonists.

Agents	Drug	Indication	Development Stage
GLP-1RA/amylin analog combination	Semaglutide/cagrilintide	Obesity, T2DM	Phase II
GLP-1/glucagon dual agonist	Cotadutide	T2DM, NASH	Phase II
	BI 456906	Obesity, NASH, T2DM	Phase II
	Efinopegdutide (^LAPS^GLP/GCG)	NASH	Phase II
GLP-1/GIP/glucagon tri-agonists	HM15211 (^LAPS^Triple Agonist)	NASH	Phase II

Abbreviations: GIP, glucose-dependent insulinotropic polypeptide; GLP-1, glucagon-like peptide 1; GLP1RA, GLP-1 receptor agonist; NASH, non-alcoholic steatohepatitis; T2DM, type 2 diabetes mellitus.

## Data Availability

Not applicable.

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
