# Peer review of "Hormonal Gut–Brain Signaling for the Treatment of Obesity"

_ijms, 2023, doi:10.3390/ijms24043384_

Round 1
Reviewer 1 Report
The brain, particularly the hypothalamus and brainstem, monitors and integrates circulating metabolic signals, including gut hormones. Gut-brain communication is also mediated by the vagus nerve, which transmits various gut-derived signals. Recent advances in our understanding of molecular gut-brain communication promote the development of next-generation anti-obesity medications that can safely achieve substantial and lasting weight loss comparable to metabolic surgery. Herein, we comprehensively review the current knowledge about the central regulation of energy homeostasis, gut hormones involved in the regulation of food intake, and clinical data on how these hormones have been applied to the development of anti-obesity drugs. Insight and understanding for gut-brain axis may provide new therapeutic perspective for treatment of obesity and diabetes.
The topic is interesting and the review is well organization. However, there still have some issue need to be cleared.
1. Line28-31, “Obesity is associated with increasing disease burden, including type 2 diabetes, cardiovascular disease, nonalcoholic steatohepatitis, chronic kidney disease, site-specific cancers, musculoskeletal disorders, malignancies, and premature death”. It should be metabolic syndrome. Please refer this paper (Oat phenolic compounds regulate metabolic syndrome in high fat diet-fed mice via gut microbiota. Food Bioscience. 50(2022)101946. Doi: 10.1016/j.fbio.2022.101946).
2. Line52-53, “The role of gut hormones and gut-brain communication in the regulation of food intake has become a hot topic in recent years.” The food intake also effect the gut microbiota and gut-brain axis(Whole grain benefit: oat β-glucan and phenolic compounds synergistically regulates hyperlipidemia via gut microbiota in high-fat-diet mice. Food & Function, 2022, 13, 12686-12696. Doi: 10.1039/d2fo01746f.).
3. The gut microbiota and the food also have effect on the obesity and diabetes (Dietary polyphenols: regulate the advanced glycation end products (AGEs)-RAGE axis and the microbiota-gut-brain axis to prevent neurodegenerative diseases. Critical Reviews in Food Science and Nutrition. Doi: 10.1080/10408398.2022.2076064.).
4. Neurotransmitters should be summarized.
5. The hormones-gut-brain axis should be graph in the manuscript.
6. The linguistic expression needs to improve.
Author Response
Comments and Suggestions for Authors
The brain, particularly the hypothalamus and brainstem, monitors and integrates circulating metabolic signals, including gut hormones. Gut-brain communication is also mediated by the vagus nerve, which transmits various gut-derived signals. Recent advances in our understanding of molecular gut-brain communication promote the development of next-generation anti-obesity medications that can safely achieve substantial and lasting weight loss comparable to metabolic surgery. Herein, we comprehensively review the current knowledge about the central regulation of energy homeostasis, gut hormones involved in the regulation of food intake, and clinical data on how these hormones have been applied to the development of anti-obesity drugs. Insight and understanding for gut-brain axis may provide new therapeutic perspective for treatment of obesity and diabetes.
The topic is interesting and the review is well organization. However, there still have some issue need to be cleared.
- Line28-31, “Obesity is associated with increasing disease burden, including type 2 diabetes, cardiovascular disease, nonalcoholic steatohepatitis, chronic kidney disease, site-specific cancers, musculoskeletal disorders, malignancies, and premature death”. It should be metabolic syndrome. Please refer this paper (Oat phenolic compounds regulate metabolic syndrome in high fat diet-fed mice via gut microbiota. Food Bioscience. 50(2022)101946. Doi: 10.1016/j.fbio.2022.101946).
Response: Based on your comments, we added 'metabolic syndrome' as a comorbidity of obesity with the reference you recommended (line 32).
- Line52-53, “The role of gut hormones and gut-brain communication in the regulation of food intake has become a hot topic in recent years.” The food intake also effect the gut microbiota and gut-brain axis(Whole grain benefit: oat β-glucan and phenolic compounds synergistically regulates hyperlipidemia via gut microbiota in high-fat-diet mice. Food & Function, 2022, 13, 12686-12696. Doi: 10.1039/d2fo01746f.).
Response: With your comments, we added the reference you recommended (line 57).
- The gut microbiota and the food also have effect on the obesity and diabetes (Dietary polyphenols: regulate the advanced glycation end products (AGEs)-RAGE axis and the microbiota-gut-brain axis to prevent neurodegenerative diseases. Critical Reviews in Food Science and Nutrition. Doi: 10.1080/10408398.2022.2076064.).
Response: According to your comments, we added the reference you recommended (line 57).
- Neurotransmitters should be summarized.
Response: We have summarized appetite-related neuropeptides as follows (line 76-79).
“Two subpopulations of ARC neurons express appetite-related neuropeptides: anorexigenic pro-opiomelanocortin (POMC) and orexigenic Agouti-related protein (AgRP) neurons. Neuropeptide Y (NPY), another potent orexigenic peptide, is co-expressed with AgRP.”
- The hormones-gut-brain axis should be graph in the manuscript.
Response: We have provided a graphical summary of the hypothalamic and brainstem neural circuits that regulate food intake and the action of gut hormones on these areas in Figure 1.
- The linguistic expression needs to improve.
Response: We entrusted English proofreading to a professional.
Reviewer 2 Report
Roh et al. provide a discussion of the current understanding of gut-brain signaling for the treatment of obesity. The review covers how the hormones that are released by different organs communicate the nervous system to control metabolism and appetite and hence obesity. I find the review interesting and well-written. Below are my comments that will help to improvise the manuscript before publication.
1. Authors should trim down some of the statistics information in the introduction.
2. The authors did not discuss serotonin and how this perturbation of this signaling can also induce obesity. Serotonin is a neurotransmitter, however, is regarded as a hormone as well.
3. Please check: In line 244, amylin activates CGRP.
Author Response
Comments and Suggestions for Authors
Roh et al. provide a discussion of the current understanding of gut-brain signaling for the treatment of obesity. The review covers how the hormones that are released by different organs communicate the nervous system to control metabolism and appetite and hence obesity. I find the review interesting and well-written. Below are my comments that will help to improvise the manuscript before publication.
- Authors should trim down some of the statistics information in the introduction.
Response: According to your comments, we moved the prevalence of diabetes in Korean adults and prevalence of obesity in these patients from the Introduction to the Main body (line 297-298).
- The authors did not discuss serotonin and how this perturbation of this signaling can also induce obesity. Serotonin is a neurotransmitter, however, is regarded as a hormone as well.
Response: Thank you for your comment. We added the anorexigenic effect of serotonin as follows (line 83-84).
“The anorexic effect of POMC neurons is mediated in part by the action of serotonin through the 5HT-2C receptor on these neurons [30].”
- Please check: In line 244, amylin activates CGRP.
Response: Thanks to your comment, we have modified the description for amylin as follows (line 256-258).
“Amylin is co-secreted with insulin from the pancreatic β-cells and highly homologous to calcitonin and calcitonin gene-related peptide (CGRP). It activates specific receptors including the calcitonin G protein-coupled receptor.”
Reviewer 3 Report
Roh and Choi discussed the importance of GLP-1 agonists and their effects in obesity treatment. The manuscript is interesting, but it would greatly benefit from the below points, to improve its quality:
1) a comparative table between GLP-1 agonists containing info of doses, administration routes, weight loss percentage for each doses, length of treatment to achieve maximum goal and maintenance schemes, etc, for overweight and obese individuals.
2) Ongoing studies on novel therapies developed based on GLP-1 targeting.
Author Response
Comments and Suggestions for Authors
Roh and Choi discussed the importance of GLP-1 agonists and their effects in obesity treatment. The manuscript is interesting, but it would greatly benefit from the below points, to improve its quality:
1) a comparative table between GLP-1 agonists containing info of doses, administration routes, weight loss percentage for each doses, length of treatment to achieve maximum goal and maintenance schemes, etc, for overweight and obese individuals.
Response: Thank you for your comments. We added a table comparing weight-loss effects of liraglutide, semaglutide, and tirzepatide (Table 1).
2) Ongoing studies on novel therapies developed based on GLP-1 targeting.
Response: Thank you for your comments. We added a table for a list of novel therapies for obesity and diabetes consisting of GLP-1 based multi-agonists (Table 2).
Reviewer 4 Report
Line 23-27: The author may add some possible reasons. Was this diet-induced, or other lifestyle change (less exercise) or genetic/epigenetic factors?
Line 42: Can you elaborate more that which therapies you are talking about? It would be great of you remain more specific.
Line 64: 20% loss in how much time, is this in line with your previous statement in line 42.
Line 84: This 6% is general for all populations or specifies certain populations.
The author can explain in one separate portion about the I or L cell of the intestine & physiological functions.
Line 266: Again author put 2 statements, “CCK1R is mediated by ghrelin” & “leptin & CCK acts synergistically”, but he did not mention, humans or animals, if humans which populations, if animals then what is its confirmation in humans? This is not very easy for the reader to go on each reference to find missing information. This is a general flaw throughout the review. This should be updated here and throughout the manuscript. If species are not mentioned, generally, it is considered that author is talking about humans, but still, a population background is very important in these statements.
Line 269: Better to mention the agonist name and dosage also and again test population & type of study is very important.
Line 272: Now author did not mention from which cell OXM is secreted.
Line 278-280: So, what was the conclusion of that study?
Line 284-285: What is the mechanism behind the together action of GLP-1R & SGLT2i?
Similarly, all clinical trials mentioned by the authors should be discussed in the populations, For example, diet-induced obesity is more common and relevant in western populations due to lifestyle. This information should be added along with the sample size of tested populations in all discussions.
Secondly, the authors repeatedly define obesity as weight loss. But weight is one factor in the calculation of obesity. It defines generally BMI, which is missing throughout the discussion of all medications. The third and most important factor was the blood lipid profiles of these patients, which are associated with cardiovascular risks in addition to T2DM.
Author Response
Comments and Suggestions for Authors
Line 23-27: The author may add some possible reasons. Was this diet-induced, or other lifestyle change (less exercise) or genetic/epigenetic factors?
Response: Based on your comments, we added dietary factors associated with significant increase in obesity, especially in younger adults (line 28-31).
Line 42: Can you elaborate more that which therapies you are talking about? It would be great of you remain more specific.
Response: Thank you for your comments. To reduce misunderstanding by readers, we changed “anti-obesity drugs” to “pharmacological management of obesity” (line 38).
Line 64: 20% loss in how much time, is this in line with your previous statement in line 42.
Response: According to your comments, we modified the previous sentence as follows.
“Overall, most approved anti-obesity medications prior to semaglutide and tirzepatide have been shown to provide a modest placebo-subtracted weight reduction of 3‒7% after 6–12 months of treatment,”
Line 84: This 6% is general for all populations or specifies certain populations.
Response: We changed the sentence more clearly as follows.
“MC4R mutations in humans are the most common cause of severe early-onset obesity (up to 6% of early-onset obesity cases before 10 years of age)”,
The author can explain in one separate portion about the I or L cell of the intestine & physiological functions.
Response: Thank you for your suggestion. We added separate portion about enteroendocrine cells as follow.
“3.1. Enteroendocrine cells
The gut is one of the largest hormone-producing organs in the human body. The enteoendocrine cells are located throughout the gastrointestinal tract and constitute only <1% of the intestinal epithelial cell population [54]. However, they have critical roles as an important component of the gut–brain axis [55]. There are at least 15 types of EEC have been described and produces more than 20 hormones that influence processes including insulin secretion, gut motility, and food intake. Previously characterized EEC families include I-cells secreting CCK, K-cells secreting GIP, and L-cells secreting GLP-1 and PYY. Recent work suggested that ECCs exhibit overlap of hormonal expression which reflect factors such as location along the gut and exposure to ingested nutrients [56].”
Line 266: Again author put 2 statements, “CCK1R is mediated by ghrelin” & “leptin & CCK acts synergistically”, but he did not mention, humans or animals, if humans which populations, if animals then what is its confirmation in humans? This is not very easy for the reader to go on each reference to find missing information. This is a general flaw throughout the review. This should be updated here and throughout the manuscript. If species are not mentioned, generally, it is considered that author is talking about humans, but still, a population background is very important in these statements.
Response: We modified that sentence more clearly as follows.
“In addition, hyperphagia in CCK1R null mice ingesting HF diet is mediated by ghrelin [108], and food-reducing effect of CCK is mediated by a functional synergistic interaction between leptin and CCK in the hypothalamic PVN [109].”
Line 269: Better to mention the agonist name and dosage also and again test population & type of study is very important.
Response: Thank you for your comment. We added a table comparing study design, study population, study intervention, and study results of representative clinical trials of liraglutide, semaglutide, and tirzepatide (Table 1).
Line 272: Now author did not mention from which cell OXM is secreted.
Response: Based on your comment, we changed the sentence as follows (line 286-288).
“OXM is secreted from the intestinal L cells together with GLP-1 and PYY in the post-prandial state and exerts its anorectic action primarily thorough binding to GLP-1R and the glucagon receptor with lower affinity [112].”
Line 278-280: So, what was the conclusion of that study?
Response: Thank you for your comment. We added the meaning of findings of that study as follow (line 292-295).
“Manganese-enhanced magnetic resonance imaging studies have shown differential neural activation between OXM and GLP-1 in the hypothalamus [117], indicating that OXM may act via different hypothalamic pathways than those of GLP-1.”
Line 284-285: What is the mechanism behind the together action of GLP-1R & SGLT2i?
Response: Thank you for your comment. We added mechanism of action of combination therapy with GLP1RA and SLGT2i (line 300-305).
“In addition, combination treatment using both GLP-1R agonists and sodium-glucose cotransporter 2 inhibitor (SGLT2i) is effective and tolerable in patients with type 2 diabetes [120]. Since SGLT2i has a different mechanism of action that inhibits glucose reabsorption in the renal proximal tubule and leads to glucosuria, combination therapy with GLP1RA and SGLT2i may produce additive metabolic and cardiovascular benefits [121].”
Similarly, all clinical trials mentioned by the authors should be discussed in the populations, For example, diet-induced obesity is more common and relevant in western populations due to lifestyle. This information should be added along with the sample size of tested populations in all discussions.
Response: Thank you for your comment. We added a table comparing study design, study population, study intervention, and study results of representative clinical trials of liraglutide, semaglutide, and tirzepatide (Table 1).
Secondly, the authors repeatedly define obesity as weight loss. But weight is one factor in the calculation of obesity. It defines generally BMI, which is missing throughout the discussion of all medications. The third and most important factor was the blood lipid profiles of these patients, which are associated with cardiovascular risks in addition to T2DM.
Response: According to your comment, we added improvement of cardiometabolic risk factors associated with treatment of all drugs as follows (line 325-327, line 360-362, line 393-394).
“Moreover, liraglutide treatment was associated with greater reductions in BMI, waist circumference, blood pressure, fasting lipid levels, and inflammatory markers than placebo.”
“Weight loss with semaglutide was accompanied by greater reductions than placebo in waist circumference, BMI, blood pressure, lipid levels, and inflammatory markers.”
“Tirzepatide treatment also resulted benefits with respect to changes in waist circumference, blood pressure, and lipid levels.”
We also added cardiovascular outcome trials of liraglutide (LEADER) and semaglutide (SUSTAIN 6) (line 332-336, line 351-352) as follows.
“Effects of liraglutide on cardiovascular outcomes was demonstrated in the LEADER trial (Liraglutide and Cardiovascular Outcomes in Type 2 Diabetes) that treatment of liraglutide 1.8mg was associated with a 13% significant relative risk reduction in major adverse cardiovascular events in patients with type 2 diabetes and high cardiovascular risk [129].”
“In SUSTAIN 6 trial, there were fewer first major adverse cardiovascular event with semaglutide with 16% relative risk reduction [133].”
Round 2
Reviewer 1 Report
The author has response the reviewer's comments. The review is interesting and well organization.
However, It is better to check the nano deliver sytem, such as nano gel or nanoparticle to help hormonal signal exert the role(Research Advances of Lactoferrin in Electrostatic Spinning, Nano Self-Assembly, and Immune and Gut Microbiota Regulation. Journal of Agricultural and Food Chemistry. 70(33):10075-10089. Research Advances of Lactoferrin in Electrostatic Spinning, Nano Self-Assembly, and Immune and Gut Microbiota Regulation. Journal of Agricultural and Food Chemistry. 70(33):10075-10089.).
Author Response
Response: Thank you for your valuable comments. Oral semaglutide has been developed as we mentioned in the manuscript as follows (line 342-344).
“Oral semaglutide has been developed as a permeation enhancer to increase solubility by increasing the pH of the local environment and using a passive transcellular route across cell membranes [131].”
We also added nano-delivery of GLP-1 as follows (line 344-345).
“In addition, oral delivery of GLP-1 through nanoparticles suggested better systemic and tissue bioavailability [132].”